# Dynamic Prompt Evolution via Multi-Attribute Feedback for Text-to-Image Generation

## Abstract

Most existing text-to-image methods primarily focus on enhancing model comprehension or tuning prompt strength, while overlooking the intrinsic expressive limitations of the prompts themselves—particularly in aligning the generated content with user-specified attributes. In this paper, we propose *Dynamic Prompt Evolution via Multi-Attribute Feedback* (DPE-MAF), a framework that integrates heuristic algorithms with the prior knowledge of large language models (LLMs) to dynamically generate optimal prompts tailored to user-intended attributes for text-to-image generation. Specifically, we formulate the prompt optimization task as a zero-order optimization problem within the natural language space. To address this, we introduce a *Diffusion-Heuristic Optimization* (DHO) module, which employs LLMs to expand an initial prompt into a candidate population and performs heuristic iterative search guided by multiple attributes, thereby dynamically steering the natural language optimization toward user-desired outputs. To further guide the iterative process, we propose a *Dynamic Contrastive Guidance Update* (DCGU) mechanism, which refines the search by contrasting semantic features between high-quality and low-quality prompts. This contrastive feedback facilitates convergence toward the global optimum. Our approach significantly enhances the ability of diffusion-based generative models to produce semantically consistent and high-fidelity images in text-to-image tasks. Extensive experiments demonstrate that DPE-MAF effectively evolves prompts to improve the performance of various diffusion models, surpassing state-of-the-art prompt-based approaches in text-to-image generation.

## 1 Introduction

The rapid advancement of Text-to-Image (T2I) diffusion models (Ho et al., 2020; Sohl-Dickstein et al., 2015) has enabled users to generate high-quality and creative images by simply inputting natural language descriptions (Ramesh et al., 2022; Rombach et al., 2022; Saharia et al., 2022). Models such as Stable Diffusion (Rombach et al., 2022) and its variant Flux (Batifol et al., 2025), as well as other diffusion-based systems including DALL-E2 (Ramesh et al., 2022) and Imagen (Saharia et al., 2022), leverage strong conditional generation capabilities to inject text embeddings into the generation process (Dhariwal & Nichol, 2021; Ho & Salimans, 2022), thereby achieving unprecedented freedom in creative expression. However, stably obtaining results with desired attributes, such as higher aesthetic quality, semantic alignment, and human-preference agreement, remains challenging. A key reason is the reliance on maximum-likelihood training (Ho et al., 2020), which misaligns with user-side rewards, causing the generated image to deviate from user's intention.

Existing enhancement strategies for T2I diffusion models fall into three lines: (1) direct fine-tuning to align with external rewards or human preferences (Black et al., 2024; Fan et al., 2023b; Prabhudesai et al., 2023; Zhang et al., 2025), which requires parameter access, high compute, and transfers poorly across models; (2) prompt-level adaptation (Clark et al., 2024; Kim et al., 2023; Wang et al., 2024; Mo et al., 2024a), which keeps the base model fixed and is naturally zero-shot transferable, but reinforcement/policy learning often over-optimizes rewards and collapses to deterministic suffixes and narrow styles (Hao et al., 2023b); and (3) noise-space search (Ma et al., 2025; Zhou et al., 2024; Oshima et al., 2025), which selects or optimizes initial noise at notable computational cost. Yet mod-

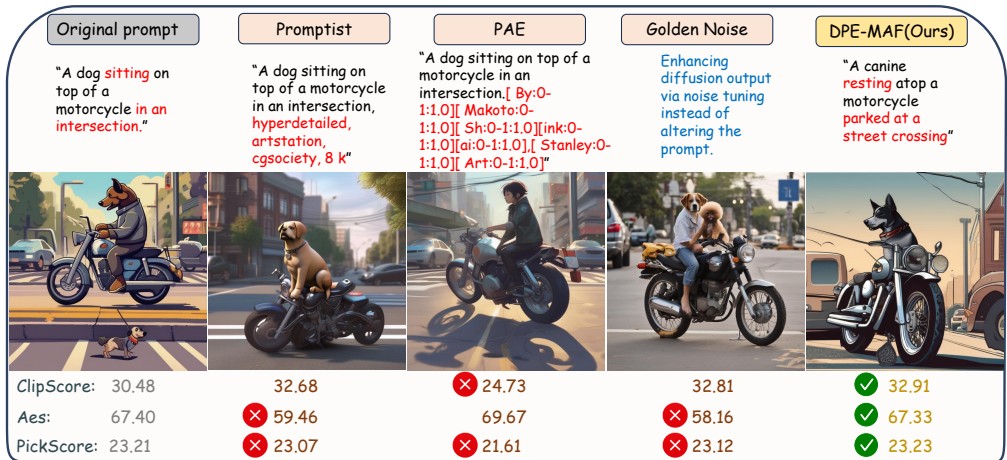

Figure 1: Comparison of methods for text-to-image generation. Compared to using the original prompt, existing methods (orange boxes) often produce variations in content and style due to suboptimal prompt optimization, leading to misalignment with user expectations. In contrast, our method yields satisfactory results with improved semantic alignment and visual aesthetics.

els frequently misread prompts: for "a dog sitting on a motorcycle at an intersection," the output may show a dog driving through an intersection. This indicates that many failures arise from suboptimal wording near the same core semantics. Therefore, given an initial prompt, style-weighted prompt expansions (PAE (Mo et al., 2024b)), noise tuning (Golden Noise (Zhou et al., 2024)), and the addition of decorative suffixes to prompts (Promptist (Hao et al., 2023b)) are usually adopted to obtain better generative results, as illustrated in Figure 1(orange box), which summarizes typical diffusion-based T2I methods. However, these methods may fail to sufficiently optimize prompts to align with user expectations, causing changes in content and style.

Moreover, prompt optimization itself faces several inherent challenges. Firstly, prompts are distributed in a high-dimensional discrete semantic space, where the generation quality after diffusion denoising can vary dramatically across different regions, making gradient-based optimization inapplicable. Secondly, as shown in Figure 2, different expressions of the same intent may lead to large variations in both aesthetic quality and semantic alignment. Thirdly, heuristic search in this space often gets stuck in local optima, resulting in little to no improvement from extra computation. These challenges motivate us to explore a novel framework that efficiently optimizes prompts in semantic space and promotes the production of high-fidelity images.

In this paper, we propose Dynamic Prompt Evolution via Multi-Attribute Feedback (DPE-MAF), which integrates heuristic algorithms with the prior knowledge of LLMs to generate optimal prompts with user-intended attributes for text-to-image generation. Specifically, DPE-MAF incorporates two key designs, i.e., Diffusion-Heuristic Optimization (DHO) and Dynamic Contrastive Guidance Update (DCGU). The former leverages LLMs to generate a population of prompts and applies the heuristic algorithm to select the most effective candidates, while the latter dynamically extracts contrastive semantic features from high- and low-quality prompts to guide prompt optimization, thereby promoting convergence toward the global optimum. As shown in Figure 1 (yellow box), DPE-MAF produces satisfactory results with better semantic alignment and aesthetics under prompt evolution with multi-attribute feedback without modifying the base diffusion model.

Our main contributions are summarized as follows:

- We propose **DPE-MAF** (Dynamic Prompt Evolution via Multi-Attribute Feedback), which integrates heuristic algorithms with the prior knowledge of large language models (LLMs) to dynamically generate optimal prompts that align with user-specified attributes in text-to-image generation.

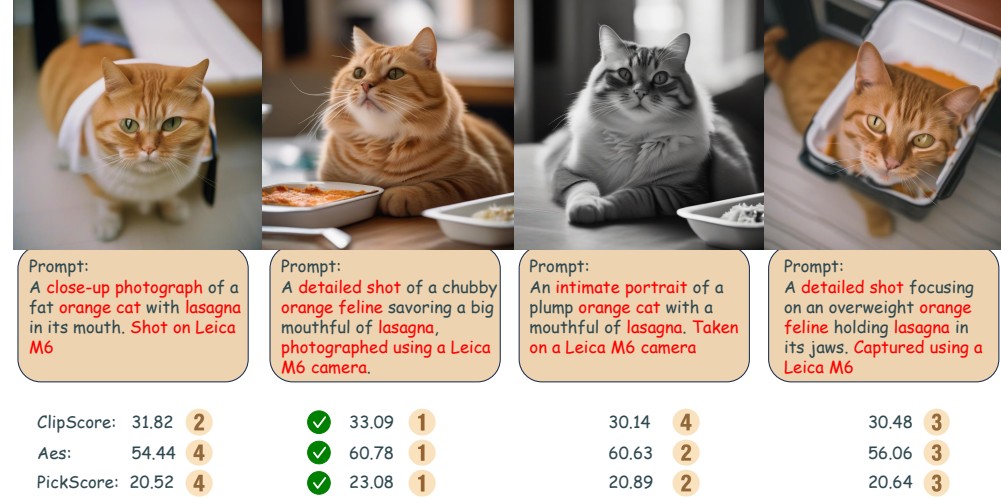

Figure 2: Different phrasings of the same meaning can lead to variations in the quality of generated images. For example, variants that specify composition and interaction ("a detailed shot ⋯ savoring a big mouthful") yield higher Aesthetic and PickScore, whereas more abstract or ambiguous wording ("an intimate portrait", "holding") weakens text–image alignment in terms of CLIPScore.

- We design a **Diffusion-Heuristic Optimization (DHO)** algorithm that expands prompts into a candidate population via LLMs, and employs heuristic search with multi-attribute feedback to iteratively refine prompts for better alignment with user intent.
- We introduce a **Dynamic Contrastive Guidance Update (DCGU)** mechanism, which leverages semantic features from both high-quality and low-quality prompts to guide the evolutionary process through a memory pool. This mitigates local stagnation and accelerates convergence toward global optima.
- Extensive experiments on multiple benchmark datasets demonstrate that our proposed DPE-MAF effectively evolves prompts and achieves superior aesthetic quality and semantic alignment.

## 2 RELATED WORK

**Diffusion Models for Text-to-Image Generation.** Diffusion models, such as DDPM(Ho et al., 2020) and DDIM (Song et al., 2021), have achieved remarkable success in the text-to-image generation domain by synthesizing high-fidelity images through iterative noise denoising processes conditioned on text prompts (Bao et al., 2023; Fan et al., 2023a; Kawar et al., 2023). Recent advances include score-based models, flow-based variants, and consistency models (Song & Ermon, 2019; Lipman et al., 2023; Liu et al., 2023; Song et al., 2023), which have substantially improved generation quality but still suffer from the semantic inconsistency between the generated images and their corresponding textual prompts.

**Large Language Models for Semantic Guidance.** Large Language Models (LLMs) have emerged as powerful tools for enhancing text-to-image generation by enabling semantic reasoning and prompt refinement (Lian et al., 2024). Integrating LLMs with diffusion models (Hao et al., 2023a; Mañas et al., 2024) helps improve image quality and alignment with textual intent. Nonetheless, ensuring accurate alignment between generated outputs and human intent remains a key challenge across a wide range of scenarios.

**Prompt Engineering in Diffusion T2I Generation.** Prompt engineering has become a critical strategy for improving image quality by optimizing the expressiveness and informativeness of input prompts. Various approaches have been proposed to restructure prompts for better guidance during the denoising process. For example, PH2P (Mahajan et al., 2024) and PAE (Mo et al., 2024b) focus on prompt refinement within the diffusion iterations, while LLM-based approaches aim to enhance semantic coherence and reduce misalignment. Reinforcement learning has also been explored, as

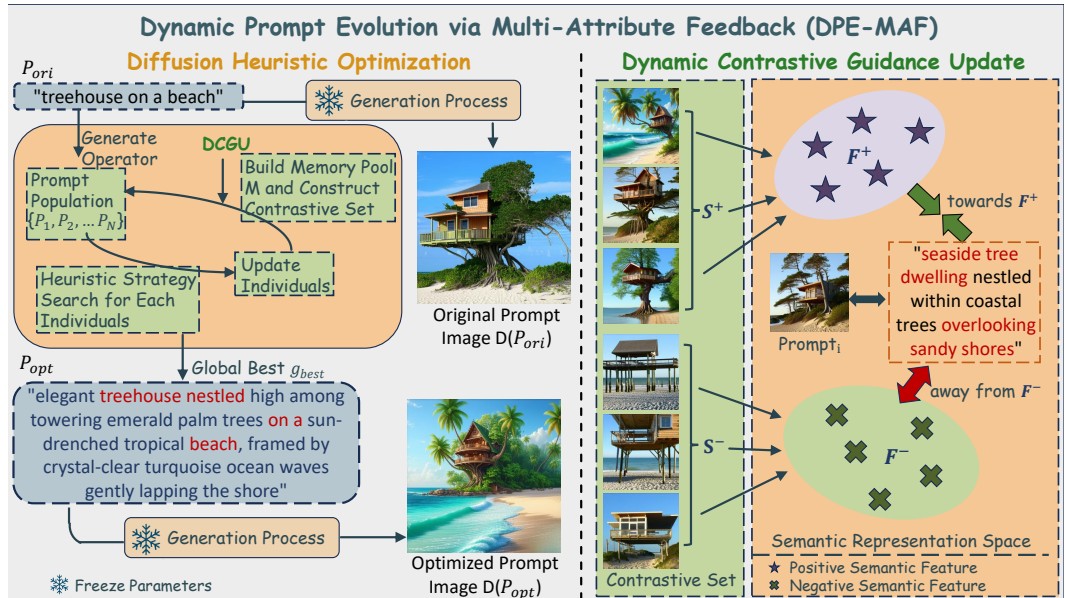

Figure 3: Overall framework of DPE-MAF. Given a user prompt, we use LLMs as generative operators to produce $N$ semantically equivalent variants ("preserve the core semantics and generate $N$ synonymous variants"). The DHO algorithm then conducts heuristic search in the natural-language space and continuously appends newly generated prompts to a memory pool $M$. At each iteration, each particle updates $P_i$ by referring to its personal best $p_{\text{best},i}$ and the global best $g_{\text{best}}$. Meanwhile, DCGU extracts contrastive features $F^+$ and $F^-$ from positive and negative prompts $S^+$ and $S^-$ in $M$ and uses them as auxiliary conditions, encouraging new prompts to inherit the strengths of $F^+$ while avoiding the weaknesses of $F^-$, thereby accelerating optimization.

in PAG (Yun et al., 2025), which formulates prompt optimization as token-level prediction using GFlowNets (Bengio et al., 2021; 2023). Despite these advances, existing methods still face difficulties with long or complex prompts and exhibit slower processing speeds under these conditions.

## 3 PROBLEM FORMULATION

Text-to-image diffusion models have achieved remarkable progress in generation quality, but generative results are highly sensitive to the quality of the input prompts. We formulate prompt optimization as the following combinatorial optimization problem:

$$P_{\text{opt}} = \arg \max_{P_i \in \mathcal{P}} S(D(P_i)), \tag{1}$$

where $P_i$ denotes a candidate prompt generated during the optimization process, $D(\cdot)$ denotes the denoising diffusion generation process, $\mathcal{P}$ is the space of natural language prompts, and $S(\cdot)$ is the composite evaluation score.

Formally, the denoising diffusion generation process can be expressed as

$$I_i = D(P_i) = \text{DiffusionSampler}(P_i, z, s), \tag{2}$$

where $z \sim \mathcal{N}(0, I)$ denotes the initial noise, and $s$ represents the random seed that controls stochasticity in the generation process. The composite evaluation score for image $I_i$ is defined as

$$f_i = \hat{\text{Clip}}(I_i) + \hat{\text{Aesthetic}}(I_i) + \hat{\text{Preference}}(I_i), \tag{3}$$

where $\hat{\text{Clip}}(\cdot)$ represents the normalized CLIPScore (Radford et al., 2021), $\hat{\text{Aesthetic}}(\cdot)$ denotes the normalized aesthetic score (Schuhmann et al., 2022), and $\hat{\text{Preference}}(\cdot)$ represents the normalized PickScore (Xu et al., 2023), which presents human preference on image quality and semantic

fidelity. Each normalization is computed as

$$\hat{s}_i = \frac{s_i - s_{\min}}{s_{\max} - s_{\min}}, \tag{4}$$

with $s_i$ being the raw score of a given metric across the current population, and $s_{\min}, s_{\max}$ the minimum and maximum values of that metric, respectively. To ensure the validity of the optimization process, the optimized prompt is required to preserve the core semantics of the initial prompt, which can be expressed as the constraint:

$$\text{sem}(P_{\text{i}}) = \text{sem}(P_{\text{init}}), \tag{5}$$

where $P_{\text{init}}$ denotes the initial prompt provided as the input to the optimization procedure, and $\text{sem}(\cdot)$ denotes the semantic representation of a prompt.

## 4 OVERVIEW OF DPE-MAF

To optimize prompts for better alignment with human-preferred attributes, we propose the Dynamic Prompt Evolution via Multi-Attribute Feedback (DPE-MAF) framework. We leverage the semantic generation and reasoning capability of large language models (LLMs) as an operator executor, enabling heuristic exploration in the natural language space. To further mitigate the risk of local optima, we introduce the Dynamic Contrastive Guidance Update (DCGU) mechanism, which adaptively extracts positive and negative semantic features from the exemplar memory pool to guide the optimization process.

### 4.1 INSTANCE OF DIFFUSION-HEURISTIC OPTIMIZATION (DHO)

The DHO component describes how heuristic optimization can be instantiated in the natural language prompt space. We present two instantiations: Particle Swarm Optimization (PSO) and Genetic Algorithm (GA). The instantiation details of GA are provided in the Appendix B.

**DHO with Particle Swarm Optimization.** In the initialization phase, a swarm of $N$ prompts $P = P_1, P_2, \ldots, P_N$ is generated and scored $S_i$. Specifically, large language models (LLMs) are employed as generative operators to execute the command: "preserve the core semantics of the input prompt and generate $N - 1$ semantically equivalent prompt variants". Each particle $P_i$ is initialized with its personal best $p_{\text{best},i} = P_i$ and $p_{\text{best},i}^{\text{score}} = S_i$. The global best $g_{\text{best}}$ is set to the highest-scoring prompt. At each iteration, under DCGU guidance, $P_i$ is first updated toward both $p_{\text{best},i}$ and $g_{\text{best}}$ to produce a candidate $P_i'$. This update is executed by large language models (LLMs), which are instructed to "maintain the original semantic meaning while moving toward both the personal best and global best prompts." To ensure the search direction and speed, the DCGU mechanism is further applied in each iteration to extract contrastive semantic features from good and bad prompts in the population as additional guidance. With probability $0.4$, a random perturbation is applied to the candidate to obtain $\tilde{P}_i = \text{LLM.PERTURB}(P_i')$; otherwise, set $\tilde{P}_i = P_i'$. If $S(D(\tilde{P}_i)) > S_i$, the update is accepted, i.e., $P_i \leftarrow \tilde{P}_i$ and $S_i \leftarrow S(D(\tilde{P}_i))$. If $S_i > p_{\text{best},i}^{\text{score}}$, then $p_{\text{best},i} \leftarrow P_i$ and $p_{\text{best},i}^{\text{score}} \leftarrow S_i$. After $G$ iterations, the global best $p_{\text{best}}$ is returned as the optimized prompt. The full procedure is detailed in Algorithm 1.

### 4.2 DYNAMIC CONTRASTIVE GUIDANCE UPDATE (DCGU)

The goal of the Dynamic Contrastive Guidance Update (DCGU) mechanism is to accelerate the heuristic search process by extracting contrastive features from historical information. DCGU consists of two components: a dynamic triggering mechanism to compute the improvement rate across generations and determine whether the threshold is reached, thereby triggering the update of guidance features, and a contrastive feature-guided update mechanism to update semantic contrastive features that guide the search toward better results.

Specifically, we firstly define a memory pool $M = \{(P_i, f_i, I_i)\}_{i=1}^{N}$ that stores triplets where $P_i$ is the prompt, $f_i$ is its normalized score, and $I_i$ is the corresponding generated image. During the search, each newly generated prompt is inserted into $M$ as $(P_i, f_i, I_i)$. An adaptive triggering

---

**Algorithm 1** PSO Instance with DCGU for Prompt Optimization

---

**Require:** Initial prompt $P_{\text{init}}$; swarm size $N$; iterations $G$; diffusion sampler $D(\cdot)$; composite score $S(\cdot)$; tolerance $\varepsilon = 10^{-6}$

**Ensure:** Optimized prompt $p_{\text{opt}}$

1: **Initialize** swarm $P = \{P_1, \dots, P_N\} \leftarrow \text{LLM.GENERATOR}(P_{\text{init}})$

2: For each $P_i$: $I_i \leftarrow D(P_i)$; $S_i \leftarrow S(I_i)$; $p_{\text{best},i} \leftarrow P_i$; $p_{\text{best},i}^{\text{score}} \leftarrow S_i$

3: $g_{\text{best}} \leftarrow \arg\max_{P_i \in P} S_i$; $\quad b_0 \leftarrow \max_i S_i$

4: Build memory pool $\mathcal{M} \leftarrow \{(P_i, f_i, I_i)\}_{i=1}^{N}$

5: Initialize features: $\mathcal{T}^+ \leftarrow \text{TOPK}(\mathcal{M}, K{=}3, \text{key} = S)$; $\mathcal{T}^- \leftarrow \text{BOTTOMK}(\mathcal{M}, K{=}3, \text{key} = S)$; $(F^+, F^-) \leftarrow \text{LLM\_EXTRACTFEATURES}(\mathcal{T}^+, \mathcal{T}^-)$

6: **for** $t = 1$ **to** $G$ **do**

7:     **DCGU trigger**: $b_t^{\text{pre}} \leftarrow \max_i S_i$; $\; r_t = \dfrac{b_t^{\text{pre}} - b_{t-1}}{\max(\varepsilon, |b_{t-1}|)}$; $\; \tau_r(t) = \tau_{r,\text{hi}} - \dfrac{\tau_{r,\text{hi}} - \tau_{r,\text{lo}}}{G-1}(t-1)$

8:     **if** $r_t < \tau_r(t)$ **then**

9:        **Update features:**

10:          $\mathcal{T}^+ \leftarrow \text{TOPK}(\mathcal{M}, K{=}3, \text{key} = S)$

11:          $\mathcal{T}^- \leftarrow \text{BOTTOMK}(\mathcal{M}, K{=}3, \text{key} = S)$

12:          $(F^+, F^-) \leftarrow \text{LLM.EXTRACTFEATURES}(\mathcal{T}^+, \mathcal{T}^-)$

13:     **else**

14:        **Keep** $(F^+, F^-)$ **unchanged**

15:     **end if**

16:     **for** each particle $P_i \in P$ **do**

17:        **Guided update:** $P_i' \leftarrow \text{LLM.UPDATE}\big(P_i, p_{\text{best},i}, g_{\text{best}} \,\big|\, F^+, F^-\big)$

18:        **if** $\text{RANDOM}() < 0.4$ **then**

19:          **Perturbation**: $P_i \leftarrow \text{LLM.PERTURB}(P_i)$

20:        **end if**

21:        **if** $S(D(P_i')) > S_i$ **then**

22:          $P_i \leftarrow P_i'$; $\quad S_i \leftarrow S(D(P_i'))$

23:        **end if**

24:        **if** $S_i > p_{\text{best},i}^{\text{score}}$ **then**

25:          $p_{\text{best},i} \leftarrow P_i$; $\quad p_{\text{best},i}^{\text{score}} \leftarrow S_i$

26:        **end if**

27:        Push $(P_i, f_i, I_i)$ into memory pool $\mathcal{M}$

28:     **end for**

29:     $g_{\text{best}} \leftarrow \arg\max_{p_{\text{best},i}} p_{\text{best},i}^{\text{score}}$; $\quad b_t \leftarrow \max_i S_i$

30: **end for**

31: $p_{\text{opt}} \leftarrow g_{\text{best}}$

32: **return** $p_{\text{opt}}$

---

mechanism is introduced to detect stagnation during optimization. Let $b_g$ denote the normalized best score at generation $g$. The relative improvement rate is defined as

$$r_g = \frac{b_g - b_{g-1}}{\max(\varepsilon, |b_{g-1}|)}, \quad \varepsilon = 10^{-6}. \tag{6}$$

We define a linearly decaying threshold as

$$\tau_r(g) = \tau_{r,\text{hi}} - \frac{\tau_{r,\text{hi}} - \tau_{r,\text{lo}}}{G-1}(g-1). \tag{7}$$

If $r_g < \tau_r(g)$, the DCGU mechanism is triggered to refresh the guidance feature for the population update. Upon activation, DCGU constructs a contrastive set by selecting the top-3 and bottom-3 prompts from the memory pool $M$ based on their scores:

$$S^+ = \text{TOPK}(\{P_i \mid (P_i, f_i, I_i) \in M\}, K{=}3), \quad S^- = \text{BOTTOMK}(\{P_i \mid (P_i, f_i, I_i) \in M\}, K{=}3). \tag{8}$$

The LLM is then prompted to analyze $S^+$ and extract positive features $F^+$, and similarly $S^-$ for negative features $F^-$. Specifically, the LLM is instructed: "For the high-scoring prompts analyze and extract their common positive features" and "For the low-scoring prompts analyze and extract their common negative features". These features are injected as additional conditions into the operator instruction:

"Create a new prompt that incorporates the positive features $[F^+]$ and avoids the negative features $[F^-]$."

This feature-based guidance is then used to steer the update process. After performing $G$ iterations of updates on each individual in the population, we obtain the optimized prompt $p_{\text{opt}}$.

Table 1: Comparison of DPE-MAF with representative baseline methods on multiple datasets. 'CLIP', 'Aes', and 'Pick' denote CLIPScore, AestheticScore, and PickScore, respectively. All results indicate the percentage (%) improvement over directly generating with SDXL.

| Method | COCO | | | Lexica | | | Pick-a-Pic | | | DiffusionDB | | |
|---|---|---|---|---|---|---|---|---|---|---|---|---|
| | CLIP↑ | Aes↑ | Pick↑ | CLIP↑ | Aes↑ | Pick↑ | CLIP↑ | Aes↑ | Pick↑ | CLIP↑ | Aes↑ | Pick↑ |
| Promptist | -2.68 | 13.35 | 0.39 | -1.99 | 6.46 | 1.06 | -1.79 | 7.33 | 1.00 | -3.59 | 6.28 | 0.58 |
| PAE | -4.67 | 11.58 | -0.80 | -1.66 | 2.02 | -0.07 | -2.54 | 3.73 | -0.23 | -2.60 | 2.61 | -0.35 |
| Golden Noise | 0.11 | -1.35 | 2.13 | 2.14 | 2.59 | 2.99 | 2.04 | 1.85 | 2.92 | 1.02 | 2.11 | 2.75 |
| **DPE-MAF (Ours)** | **6.56** | 10.40 | **7.42** | **8.68** | **12.00** | **8.32** | **8.99** | **11.16** | **10.49** | **7.66** | **11.01** | **8.32** |

detailed cinematic moody colors studio portrait of a victorian young lady sleeping in a victorian pond

white mech arc - angel cyclops

a beautiful portrait of pikachu

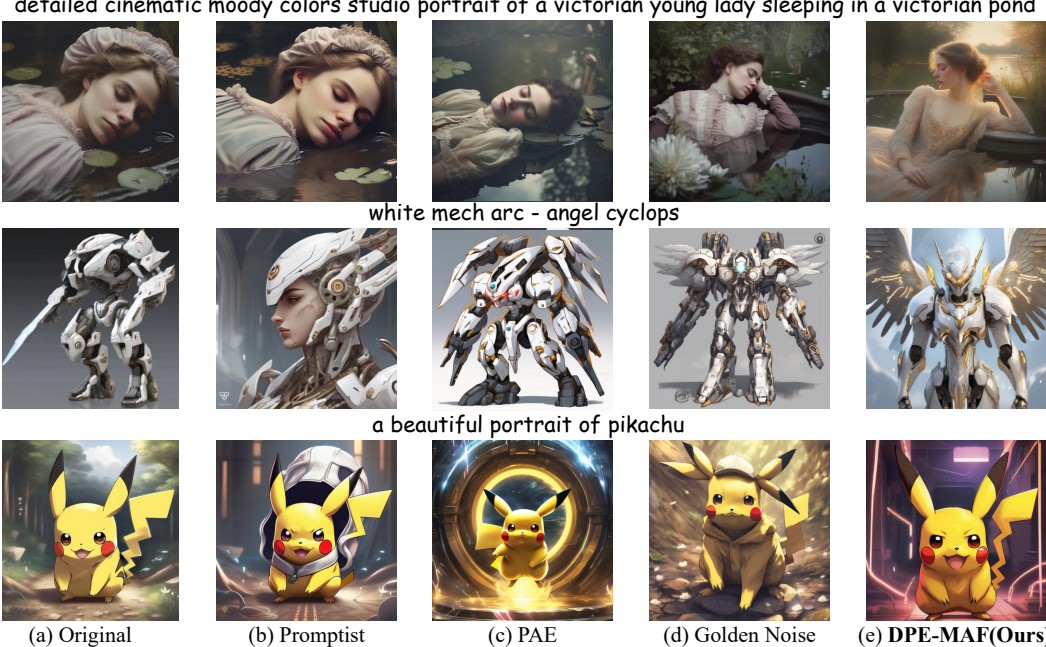

(a) Original    (b) Promptist    (c) PAE    (d) Golden Noise    (e) **DPE-MAF(Ours)**

Figure 4: Visual comparison of various methods on DiffusionDB (Wang et al., 2023) dataset. Our DPE-MAF is able to produce satisfactory results with better semantic alignment and aesthetics.

## 5 EXPERIMENTS

### 5.1 EXPERIMENTAL SETUP

**Datasets.** We conduct experiments on four text-to-image datasets including DiffusionDB (Wang et al., 2023), COCO (Lin et al., 2014), Lexica (Shen et al., 2024), and Pick-a-Pic (Kirstain et al., 2023), where 1k prompts are randomly selected as experimental samples. For our DPE-MAF, we randomly select 100 prompts from the same dataset for evaluation.

**Implementation Details.** All diffusion models used in our experiments share a fixed random seed of 42. For SD1.4 (Rombach et al., 2022), SD1.5, SD2.1, and SDXL (Podell et al., 2024), we adopt dpmsolver (Lu et al., 2022) as an accelerator, perform 50 sampling steps with DDIM (Song et al., 2021) to maintain generation quality, and employ the xformers plugin for further acceleration. For SD3 and SD3.5, we use the default 28 sampling steps. Regarding hyperparameter selection in the DHO algorithm, we employ grid search (details in Appendix D.1) to determine suitable parameters balancing performance and computational cost. Specifically, we initialize 6 particles and iterate over 12 rounds.

**Baseline Methods.** We compare our method with three text-to-image generation methods, including Promptist (Hao et al., 2023b), PAE (Mo et al., 2024b), and Golden Noise (Zhou et al., 2024).

Table 2: Performance of Particle Swarm Optimization (PSO) across various datasets and different Stable Diffusion models. Shaded cells mark the best score on each dataset and model.

| Model | Pick-a-Pic | | | COCO | | | Lexica | | | DiffusionDB | | |
|---|---|---|---|---|---|---|---|---|---|---|---|---|
| | CLIP↑ | Aes↑ | Pick↑ | CLIP↑ | Aes↑ | Pick↑ | CLIP↑ | Aes↑ | Pick↑ | CLIP↑ | Aes↑ | Pick↑ |
| SD1.4 | **6.33** | 9.66 | 4.97 | **9.23** | 8.77 | 4.01 | **6.71** | **10.03** | **7.21** | **7.73** | **10.02** | **6.71** |
| SD1.5 | 5.10 | 6.79 | **5.41** | 5.07 | 5.84 | **4.07** | 4.36 | 6.05 | 4.93 | 3.32 | 4.52 | 5.11 |
| SD2.1 | 5.50 | 5.72 | 4.24 | 1.28 | 5.03 | 3.68 | 5.38 | 5.05 | 4.71 | 6.23 | 5.05 | 5.61 |
| SDXL | 2.84 | 10.14 | 4.89 | 0.65 | **9.46** | 3.36 | 0.48 | 7.11 | 4.21 | 1.75 | 6.51 | 4.02 |
| SD3 | 4.13 | 7.98 | 3.11 | 3.24 | 2.74 | 2.14 | 4.86 | 6.37 | 4.66 | 4.35 | 6.57 | 4.02 |
| SD3.5 | 2.33 | **10.74** | 5.35 | 3.99 | 6.64 | 2.13 | 4.00 | 6.49 | 6.14 | 2.95 | 7.45 | 4.22 |

## 5.2 Main Results

Table 1 presents the quantitative results on four datasets. Compared with the other approaches, our PE-MAF achieves the highest values across all three metrics. In particular, the significant improvement in PickScore provides compelling evidence that our method generates images more aligned with human preferences. Compared to the initial prompt, Promptist (Hao et al., 2023b) and PAE (Mo et al., 2024b) achieve significant improvements on AestheticScore, but their CLIPScore drop significantly due to a lack of semantic alignment. Furthermore, Figure 4 presents visual comparisons on DiffusionDB (Wang et al., 2023) dataset. Notably, Promptist appends descriptive modifiers to the input prompt, tend to produce fixed modifiers and thus homogenize the output style. PAE augments the prompt with tokens indicating denoising range and strength, often yields semantically vacuous modifiers under constrained by the language model. And Golden Noise (Zhou et al., 2024) cannot reliably guarantee the desired attributes. In contrast, our DPE-MAF framework produces images that are not only more aesthetically pleasing but also better satisfy the user-specified attributes, which benefits from our proposed DHO algorithm and DCGU mechanism. More results are shown in the Appendix E.

## 5.3 Ablation Studies

We conduct systematic ablation experiments on the key designs of the proposed DPE-MAF. All experiments are conducted on the DiffusionDB dataset and 30 randomly selected prompts, except for verifying the performance of the DHO on four datasets (Lexica, DiffusionDB, COCO, and Pick-a-Pic) and different versions of the diffusion model. More results are shown in the Appendix D.

### 5.3.1 The Effect of DHO Instantiation

To validate the transferability of the DPE-MAF framework, we select 100 samples from each of four datasets, including Lexica, DiffusionDB, COCO, and Pick-a-Pic, to comprehensively evaluate the DHO algorithm on various versions of Stable Diffusion models (SD1.4, SD1.5, SD2.1, SDXL, SD3, and SD3.5). As shown in Table 2, our DPE-MAF framework consistently achieves effective optimization across different diffusion model versions and is capable of simultaneously enhancing multiple image attributes. Moreover, our approach not only achieves significant improvements on SD1.4 but also achieves robust improvements across other versions of Stable Diffusion.

### 5.3.2 The Impact of DCGU Dynamic Threshold

We first experiment with different triggering periods of the DCGU mechanism and find that overly frequent updates of guidance features cause the optimization process to rely excessively on real-time feedback, leading to unstable convergence and reducing the extent of exploration in the natural language space, while overly sparse updates reduce search efficiency. Therefore, to determine a more appropriate triggering strategy for updating guidance features, we adopt a dynamic threshold-based activation mechanism that monitors the improvement before each iteration and automatically triggers DCGU. Table 3 shows the total gain for different settings of the threshold parameters, with the best performance achieved when $\tau_{r,\text{hi}} = 0.017$ and $\tau_{r,\text{lo}} = 0.005$.

Table 3: Ablation study of the DCGU mechanism with different threshold settings. We report results for various combinations of $\tau_{r,hi}$ and $\tau_{r,lo}$, using total gain (%) as the metric.

| | ID1 | ID2 | ID3 | ID4 | ID5 | **ID6** | ID7 | ID8 | ID9 | ID10 | ID11 | ID12 | ID13 | ID14 | ID15 | ID16 |
|---|---|---|---|---|---|---|---|---|---|---|---|---|---|---|---|---|
| $\tau_{r,hi}$ | 0.013 | 0.013 | 0.010 | 0.017 | 0.013 | **0.017** | 0.013 | 0.017 | 0.010 | 0.020 | 0.020 | 0.013 | 0.020 | 0.020 | 0.020 | 0.017 |
| $\tau_{r,lo}$ | 0.008 | 0.006 | 0.006 | 0.008 | 0.005 | **0.005** | 0.003 | 0.003 | 0.003 | 0.006 | 0.008 | 0.008 | 0.010 | 0.008 | 0.003 | 0.005 |
| **Total** ↑ | 14.55 | 15.01 | 15.00 | 15.88 | 19.00 | **23.46** | 15.33 | 15.60 | 13.73 | 13.97 | 14.50 | 15.55 | 14.55 | 14.55 | 16.99 | 16.38 |

Table 4: Ablation studies about memory pool and perturbation mechanism, as well as LLM–DHO comparison. 'CLIP', 'Aes', and 'Pick' denote CLIPScore, AestheticScore, and PickScore, respectively.

(a) Effect of memory pool design (Particle Swarm).

| Option | CLIP↑ | Aes↑ | Pick↑ | Total↑ |
|---|---|---|---|---|
| Category-Norm | **3.39** | 6.29 | 6.58 | 16.27 |
| **Aggregate-Norm** | 3.05 | **6.87** | **7.37** | **17.28** |

(b) Effect of perturbation position (Particle Swarm).

| Option | CLIP↑ | Aes↑ | Pick↑ | Total↑ |
|---|---|---|---|---|
| No perturbation | 3.17 | 6.33 | 6.71 | 16.22 |
| **Perturbation in update** | 2.50 | **7.04** | 6.64 | 16.17 |

(c) Effect of perturbation (Particle Swarm).

| Option | CLIP↑ | Aes↑ | Pick↑ | Total↑ |
|---|---|---|---|---|
| Perturbation-20% | 2.85 | 7.13 | 4.38 | 14.36 |
| Perturbation-30% | 1.77 | **7.18** | 7.32 | 16.27 |
| **Perturbation-40%** | **3.46** | 5.91 | **7.14** | **16.52** |

(d) Comparison of LLM and DHO.

| Metric | LLM | DHO (Ours) |
|---|---|---|
| **CLIP**↑ | -4.67 | **7.66** |
| **Aes**↑ | 3.71 | **11.01** |
| **Pick**↑ | 0.85 | **8.32** |

### 5.3.3 THE EFFECT OF MEMORY POOL AND PERTURBATION MECHANISM

We further examine the design of the memory pool and the role of random perturbations. A properly designed memory pool effectively improves sample utilization. As shown in Table 4(a), directly extracting features from high- and low-performing prompts and injecting them into the subsequent optimization process yields better performance than separately extracting features based on CLIP-Score, Aesthetic, and PickScore. For the perturbation mechanism, both the position of perturbation insertion and the perturbation probability have a significant impact. As depicted in Table 4(b), overly frequent perturbations disrupt promising search directions, while overly weak perturbations reduce the diversity of generated prompts. Table 4(c) demonstrates that the perturbation with 40% frequency maximizes performance gains.

### 5.3.4 COMPARISON OF LLM AND DHO

Table 4(d) reports the comparison of the prompt optimization performance by using the DPE-MAF framework with that of directly using LLMs. We observe that the prompts optimized by LLMs directly are often semantically incomplete and tend to sacrifice semantic consistency while enhancing certain attributes. In contrast, the prompts optimized by our DPE-MAF achieve significant improvements across multiple attributes.

## 6 CONCLUSION

In this work, we propose DPE-MAF, a dynamic prompt evolution via multi-attribute feedback specifically designed for text-to-image generation. Our approach formulates the prompt optimization problem as a zero-order optimization problem in the natural language space and introduces a diffusion-heuristic optimization algorithm, which enables a large language model (LLM) to perform a multi-attribute-guided heuristic iterative search to generate optimal prompts with the user's desired attributes. To effectively guide prompt optimization, we develop a dynamic contrast-guided update mechanism that guides the search process by comparing the semantic features of high-quality and low-quality prompts, thereby promoting convergence to the global optimum. Extensive experiments demonstrate that DPE-MAF outperforms state-of-the-art T2I methods, achieving superior visual quality, semantic consistency, and aesthetics across various datasets.

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

## A  LLM USAGE

The contributions of the Large Language Models (LLMs) were solely focused on improving the linguistic quality of the paper, with no involvement in the scientific content or data analysis.

## B  DHO WITH GENETIC ALGORITHM

We first initialize a population of $N$ prompts $P = \{P_1, P_2, \ldots, P_N\}$ using a generation operator, and compute the fitness $f_i$ for each prompt. In the selection phase, two parents $P_a, P_b$ are chosen based on fitness. During crossover, the DCGU mechanism is applied to guide the LLM in semantically combining $P_a$ and $P_b$ to generate a child $C$. In the mutation phase, the LLM is instructed to mutate $C$ with a $40\%$ probability of random perturbation (e.g., synonym substitution, word deletion, or paraphrasing). This process is repeated to generate $k$ offspring, which are merged with the parent population. The top-$N$ individuals are selected for the next generation. This iterative process continues until termination.

---

**Algorithm 2** GA Instance with DCGU for Prompt Optimization

---

**Require:** Initial prompt $P_{\text{init}}$; population size $N$; offspring per generation $k$; generations $T$; diffusion sampler $D(\cdot)$; composite score $S(\cdot)$; tolerance $\varepsilon = 10^{-6}$
**Ensure:** Optimized prompt $p_{\text{opt}}$
1: **Initialize** population $P = \{P_1, \ldots, P_N\} \leftarrow$ LLM.GENERATOR$(P_{\text{init}})$
2: For each $P_i$: $I_i \leftarrow D(P_i)$; $S_i \leftarrow S(I_i)$
3: Build memory pool $\mathcal{M} \leftarrow \{(P_i, S_i, I_i)\}_{i=1}^N$
4: Initialize features: $\mathcal{T}^+ \leftarrow$ TOPK$(\mathcal{M}, K{=}3, \text{key} = S)$; $\mathcal{T}^- \leftarrow$ BOTTOMK$(\mathcal{M}, K{=}3, \text{key} = S)$;
   $(F^+, F^-) \leftarrow$ LLM_EXTRACTFEATURES$(\mathcal{T}^+, \mathcal{T}^-)$
5: $b_0 \leftarrow \max_i S_i$
6: **for** $t = 1$ **to** $T$ **do**
7:    **DCGU trigger**: $b_t^{\text{pre}} \leftarrow \max_i S_i$; $r_t = \dfrac{b_t^{\text{pre}} - b_{t-1}}{\max(\varepsilon, |b_{t-1}|)}$; $\tau_r(t) = \tau_{r,\text{hi}} - \dfrac{\tau_{r,\text{hi}} - \tau_{r,\text{lo}}}{T - 1}(t - 1)$
8:    **if** $r_t < \tau_r(t)$ **then**
9:      **Update features:**
10:       $\mathcal{T}^+ \leftarrow$ TOPK$(\mathcal{M}, K{=}3, \text{key} = S)$
11:       $\mathcal{T}^- \leftarrow$ BOTTOMK$(\mathcal{M}, K{=}3, \text{key} = S)$
12:       $(F^+, F^-) \leftarrow$ LLM.EXTRACTFEATURES$(\mathcal{T}^+, \mathcal{T}^-)$
13:    **else**
14:      **Keep** $(F^+, F^-)$ **unchanged**
15:    **end if**
16:    $O \leftarrow \varnothing$ {offspring set}
17:    **for** $j = 1$ **to** $k$ **do**
18:      **Parent selection**: sample $P_a, P_b$ from $P$ by roulette wheel based on $\{S_i\}$
19:      **Crossover**: $C \leftarrow$ LLM.CROSSOVER$(P_a, P_b \mid F^+, F^-)$
20:      **Mutation**: $C \leftarrow$ LLM.MUTATE$(C \mid F^+, F^-)$
21:      **if** RANDOM$() < 0.2$ **then**
22:       **Perturbation**: $C \leftarrow$ LLM.PERTURB$(C)$
23:      **end if**
24:      **Evaluate**: $I_C \leftarrow D(C)$; $S_C \leftarrow S(I_C)$
25:      Push $(C, S_C, I_C)$ into memory pool $\mathcal{M}$; $O \leftarrow O \cup \{C\}$
26:    **end for**
27:    **Environmental selection**: $P \leftarrow$ TOPN$(P \cup O, N; \text{key} = S(D(\cdot)))$
28:    $b_t \leftarrow \max_{P_i \in P} S(D(P_i))$
29: **end for**
30: $p_{\text{opt}} \leftarrow \arg\max_{P_i \in P} S(D(P_i))$
31: **return** $p_{\text{opt}}$

---

## C  PERFORMANCE COMPARISON OF GA ON DIFFERENT SD MODELS

In this section, we present the experimental results of the GA instance across different Stable Diffusion versions and datasets. As shown in Table 5, GA demonstrates positive effects across various models and datasets, especially in notably in aesthetic quality in terms of AestheticScore and

Table 5: Performance of the Genetic Algorithm (GA) across datasets and Stable Diffusion models. Shaded cells mark the best GA score within each dataset group.

| Model | Pick-a-Pic | | | COCO | | | Lexica | | | DiffusionDB | | |
|---|---|---|---|---|---|---|---|---|---|---|---|---|
| | CLIP↑ | Aes↑ | Pick↑ | CLIP↑ | Aes↑ | Pick↑ | CLIP↑ | Aes↑ | Pick↑ | CLIP↑ | Aes↑ | Pick↑ |
| SD1.4 | 3.63 | **9.40** | **5.63** | **3.89** | 5.09 | 2.26 | 0.68 | **7.40** | 4.73 | 2.48 | 2.84 | 1.61 |
| SD1.5 | **5.61** | 7.73 | 3.58 | 2.92 | 6.15 | 2.57 | 2.80 | 5.63 | 3.04 | **3.22** | 6.21 | 3.34 |
| SD2.1 | 2.66 | 5.95 | 3.62 | 2.04 | 6.31 | 2.15 | 1.23 | 4.91 | 2.24 | 3.04 | 3.93 | 3.44 |
| SDXL | 2.40 | 8.32 | 5.57 | 1.27 | **8.48** | 2.22 | 1.02 | 6.24 | 4.28 | -0.64 | 5.57 | 4.21 |
| SD3 | 3.49 | 7.32 | 3.08 | 1.99 | 3.53 | **3.08** | **3.68** | 6.30 | 4.36 | 2.53 | 5.73 | 3.62 |
| SD3.5 | 2.78 | 8.09 | 3.33 | 2.22 | 5.01 | 2.17 | 2.40 | 6.51 | **5.62** | 1.91 | **7.19** | **4.28** |

PickScore. For semantic alignment, although SDXL on DiffusionDB shows no CLIPScore gain, other models (e.g., SD1.5 gains 5.61% on Pick-a-Pic and 3.22% on DiffusionDB) achieve high scores, demonstrating the GA's strong cross-model transferability and its ability to preserve semantic consistency across datasets.

# D  ADDITIONAL EXPERIMENTS

## D.1  THE EFFECT OF HYPERPARAMETER GRID SEARCH

We conduct experiments with grid search to explore the effects of hyperparameter settings, as shown in Table 6. The results show that the particle swarm instance achieves the most significant improvement under the configuration of Iterations $G = 12$ and Particles $N = 6$. Compared with smaller parameter settings, this configuration brings larger gains, while for larger settings such as Iterations $G = 15$ with Particles $N = 6$ or Iterations $G = 15$ with Particles $N = 8$, this configuration not only delivers better performance improvements but also requires fewer computational resources. This is because the particle swarm instance tends to perform wide and deep searches, but once sufficient computational resources are allocated, the additional benefits from further searching become limited.

## D.2  THE EFFECT OF DCGU ACTIVATION PERIOD

Table 7 presents the effect of the triggering frequency of DCGU. We find that the update cycle brings significant improvements in terms of AestheticScore and PickScore, compared to without DCGU (Table 7(a)). However, if the update cycle is too frequent, the features are updated excessively, causing the search direction to change constantly and leading to slow improvements in semantic consistency. On the other hand, if the updates are too infrequent, the search becomes insufficient and the diversity of prompts is not fully expanded, which limits improvements such as aesthetic scores. In contrast, our DCGU with adaptive stagnation (Table 7(e)) dynamically updates the guidance features based on a threshold, enabling deeper exploration of the search space while preserving semantic fidelity and yielding the best performance.

## D.3  THE EFFECT OF MEMORY POOL AND PERTURBATION MECHANISM FOR GA

Table 8 and Table 9 report the impact on memory pool configuration, perturbation probability, and perturbation position based on the genetic instance, respectively. We observe that configuring the memory pool to extract features by category of the target attributes provides more noticeable improvements in guiding the search process. In addition, introducing a 20% probability of random perturbation during the mutation operation substantially enriches the exploration of the natural language space, which in turn contributes significantly to improvements in aesthetic metrics and overall performance gains. Moreover, placing the perturbation operation inside the mutation step introduces additional randomness into prompt generation, leading to better aesthetic scores as well as improved semantic consistency.

Table 6: Hyperparameter grid search for PSO Instance with DCGU on prompt optimization (transposed). We vary PSO iterations ($G$) and swarm size ($N$) as well as report *total gain* (%).

| Option | ID1 | ID2 | ID3 | ID4 | ID5 | ID6 | ID7 | ID8 | ID9 | ID10 | ID11 | ID12 | ID13 | **ID14** | ID15 | ID16 | ID17 | ID18 |
|---|---|---|---|---|---|---|---|---|---|---|---|---|---|---|---|---|---|---|
| **Iterations** $G$ | 9 | 9 | 9 | 12 | 12 | 12 | 15 | 15 | 15 | 4 | 6 | 8 | 4 | **6** | 8 | 4 | 6 | 8 |
| **Particles** $N$ | 4 | 6 | 8 | 4 | 6 | 8 | 4 | 6 | 8 | 9 | 9 | 9 | 12 | **12** | 12 | 15 | 15 | 15 |
| **Total** ↑ | 13.40 | 14.00 | 15.68 | 13.29 | 13.82 | 14.91 | 14.78 | 14.16 | 13.54 | 14.58 | 15.32 | 14.50 | 14.68 | **16.06** | 13.30 | 14.89 | 16.24 | 13.15 |

Table 7: Effect of DCGU Activation Period. 'CLIP', 'Aes', and 'Pick' denote CLIPScore, AestheticScore, and PickScore, respectively.

| DCGU Mechanism | CLIPScore↑ | Aes↑ | PickScore↑ | Total↑ |
|---|---|---|---|---|
| (a) once DCGU | **4.20** | 6.12 | 6.05 | 16.37 |
| (b) Every Generation | 2.84 | 6.77 | 6.53 | 16.15 |
| (c) Every Two Generations | 3.09 | 6.27 | 6.99 | 16.35 |
| (d) Every Three Generations | 2.40 | 7.05 | 6.87 | 16.32 |
| **(e) Adaptive Stagnation (Ours)** | 1.79 | **7.05** | **7.88** | **16.71** |

Table 8: Effect of memory pool scoring mechanism (GA). 'CLIP', 'Aes', and 'Pick' denote CLIPScore, AestheticScore, and PickScore, respectively.

| Option | CLIP↑ | Aes↑ | Pick↑ | Total↑ |
|---|---|---|---|---|
| (a) Aggregate-Norm | 2.18 | 4.92 | 1.13 | 8.23 |
| **(b) Category-Norm (Ours)** | **2.58** | **5.77** | **1.25** | **9.60** |

Table 9: Ablation studies of the perturbation mechanism for the Genetic Algorithm (GA). 'CLIP', 'Aes', and 'Pick' denote CLIPScore, AestheticScore, and PickScore, respectively.

(a) Effect of perturbation position (GA).

| Option | CLIP↑ | Aes↑ | Pick↑ | Total↑ |
|---|---|---|---|---|
| No perturbation | 2.40 | 4.82 | 1.34 | 8.56 |
| **Perturbation in mutation** | **2.63** | **5.15** | **1.50** | **9.28** |
| Perturbation after mutation | 2.13 | 5.01 | 1.23 | 8.38 |

(b) Effect of perturbation probability (GA).

| Option | CLIP↑ | Aes↑ | Pick↑ | Total↑ |
|---|---|---|---|---|
| **Perturbation-20%** | 2.55 | **5.34** | **1.51** | **9.40** |
| Perturbation-30% | 2.39 | 4.81 | 1.31 | 8.52 |
| Perturbation-40% | **2.68** | 4.77 | 1.45 | 8.83 |

# E MORE EXPERIMENT RESULTS

We provide more visualizations of various text-to-image generation methods on four datasets. Figure 5 shows more visual comparison results on Pick-a-Pic (Kirstain et al., 2023) dataset. Figure 6 shows more visual comparison results on COCO (Lin et al., 2014) dataset. Figure 7 shows more visual comparison results on Lexica (Shen et al., 2024) dataset. Figure 8 shows more visual comparison results on DiffusionDB (Wang et al., 2023) dataset.

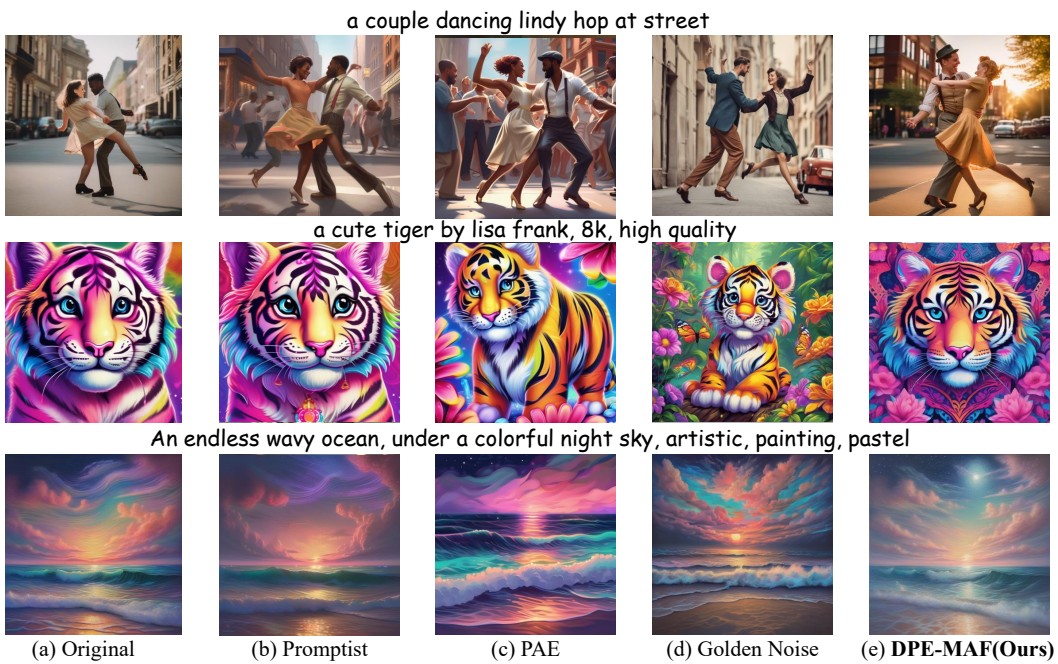

Figure 5: Visual comparison of various methods on the Pick-a-Pic (Kirstain et al., 2023) dataset. DPE-MAF demonstrates superior results in terms of semantic consistency and aesthetics.

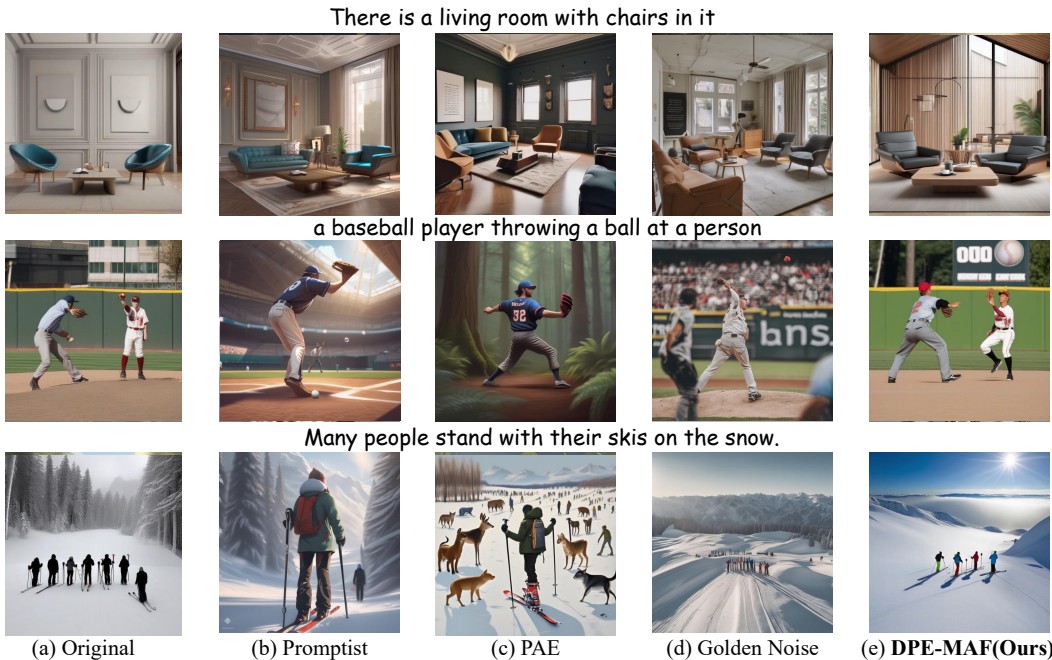

Figure 6: Visual comparison of various methods on the COCO (Lin et al., 2014) dataset. Our DPE-MAF produces images with improved semantic alignment and aesthetics.

commission of a robot chasing thugs

venom as charizard

a beautiful cute girl wearing modern stylish costume in the style of Assamese bihu mekhela sador gamosa design

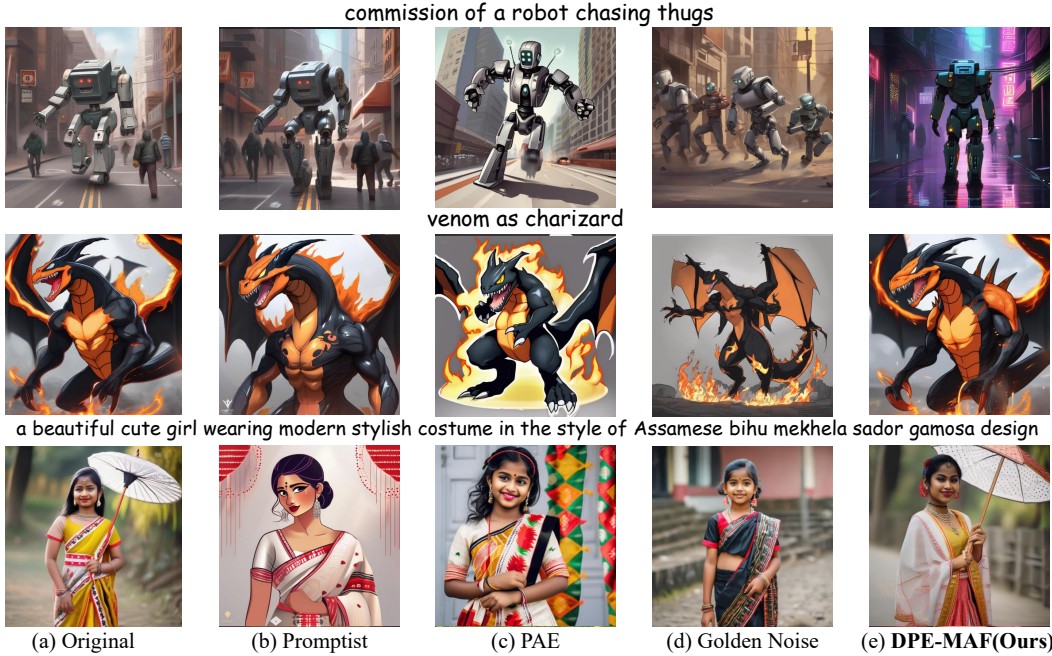

(a) Original      (b) Promptist      (c) PAE      (d) Golden Noise      (e) **DPE-MAF(Ours)**

Figure 7: Visual comparison of various methods on the Lexica (Shen et al., 2024) dataset. Our DPE-MAF enhances both semantic fidelity and aesthetic appeal.

the castle by kafka by vrubel

classic painting sonic the hedgehog standing on a tree fort dressed as a sailor pointing at a boat that is crossing the ocean

the concept of clothing of the future

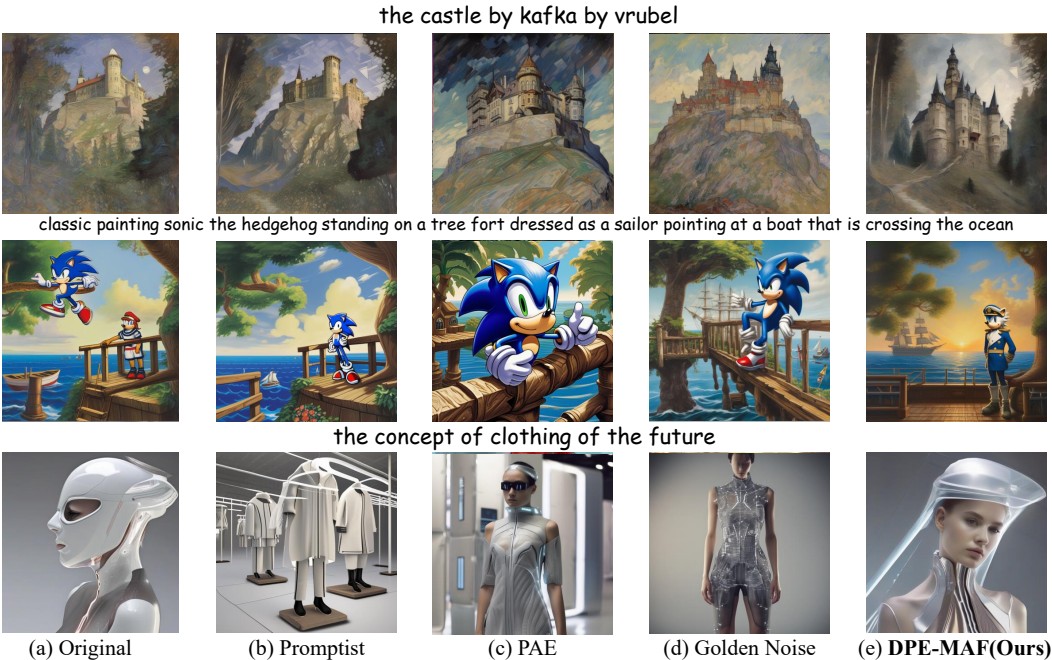

(a) Original      (b) Promptist      (c) PAE      (d) Golden Noise      (e) **DPE-MAF(Ours)**

Figure 8: Visual comparison of various methods on the DiffusionDB (Wang et al., 2023) dataset. Our DPE-MAF consistently improves both alignment and visual quality.

