# OpenReview forum: "Dynamic Prompt Evolution via Multi-Attribute Feedback for Text-to-Image Generation"
_ICLR.cc/2026/Conference — ICLR 2026 Conference Withdrawn Submission_

### Official Review · Reviewer_nbkX · 2025-10-30

**Soundness:** 2
**Presentation:** 2
**Contribution:** 2
**Rating:** 4
**Confidence:** 4

**Summary:**

The paper introduces Dynamic Prompt Evolution via Multi-Attribute Feedback, which treats prompt optimization for text-to-image generation as a zero-order optimization problem in natural language space. It integrates two main modules: (1) Diffusion-Heuristic Optimization (DHO), which employs large language models to generate semantically equivalent prompt populations and iteratively refine them using heuristic algorithms; and (2) Dynamic Contrastive Guidance Update (DCGU), which dynamically extracts contrastive features from high and low-quality prompts to guide convergence. Extensive experiments on multiple benchmark datasets show that the proposed method effectively evolves prompts and achieves superior aesthetic quality and semantic alignment.

**Strengths:**

1. The motivation is clear. The DHO and DCGU modules are independently described and visually illustrated, improving clarity of implementation.
2. The experiments show that the propose method ourperform the existing methods in CLIP, Aesthetic, and Pick scores.

**Weaknesses:**

1. Lack of ablation studies on the evaluation score design. The score in equation3 has three parts, also all of them are normalized, it is still unclear how these parts and their different combinations affect the performance.
2. Equation5 means that the semantic representation of the original prompt and the candidate prompt should be equal, how do you quantify this metric? What is the specific form of the function sem(x)? It is unclear how LLMs ensure the semantic preservation under random perturbations.
3. Lack of many details in experiments. There are no details about specific prompts, workflows, and LLMs used in the framework, which are very important for the whole procedure. It is unclear if different LLMs and instructions will affect the performance a lot.
4. Lack of efficiency analysis in the paper. For example, the inference time for each prompt.

**Questions:**

1. Typo: too long sentence in Line322.
2. What is the computation cost for the whole framework?

---

### Official Review · Reviewer_VzGh · 2025-10-31

**Soundness:** 3
**Presentation:** 3
**Contribution:** 2
**Rating:** 4
**Confidence:** 4

**Summary:**

This paper presents a novel approach for prompt optimization in text-to-image generation. The core contribution is a heuristic search algorithm that iteratively refines prompts using a Large Language Model, guided by feedback from multiple reward models assessing aesthetic quality, text-image alignment and human preference. A key innovation is the use of contrastive positive and negative examples during this evolutionary process to steer the LLM more effectively toward high-reward prompts. The work demonstrates significant potential in addressing the practical challenge of prompt engineering.

**Strengths:**

- The paper successfully automates prompt crafting by formulating it as a search problem, intelligently leveraging an LLM as a generator and reward models to provide feedbacks.

- The incorporation of positive and negative prompt examples provides crucial directional guidance to the LLM, significantly improving the efficiency and convergence of the search process compared to a naive exploration.

**Weaknesses:**

- The computational cost and time latency of the proposed heuristic algorithm are not discussed. Given that each iteration requires multiple calls to LLM, T2I models, and reward models, the total cost would be prohibitive for any real-world application that requires timely feedback.

- Methodological concerns:
    - Since different initial noises for the same prompt can lead to vastly different generated images, assessing a prompt's quality based on a single generation is unreliable and introduces substantial variance into the optimization process.
    - Given the known issue of LLM hallucination, there is no guarantee that the semantic representation of the original prompt is preserved throughout the evolutionary process, which undermines the trustworthiness of the final output.

**Questions:**

- Please provide quantitative data on the time and computational cost required for your optimization process.

- How do you address the potential variance in reward scores caused by the stochastic nature of the image generation process?

- Is relying solely on LLM instructions a sufficient guarantee for core semantic preservation?

---

### Official Review · Reviewer_B1JQ · 2025-10-31

**Soundness:** 3
**Presentation:** 3
**Contribution:** 3
**Rating:** 4
**Confidence:** 4

**Summary:**

This paper introduces the Dynamic Prompt Evolution Framework (DPE-MAF), designed to enhance semantic consistency and visual quality in text-to-image generation. To address the limitations of existing methods in prompt expressiveness and attribute alignment, DPE-MAF integrates the prior knowledge of Large Language Models (LLMs) with heuristic algorithms through the following designs: 1) The Diffusion-Heuristic Optimization (DHO) module utilizes an LLM to generate candidate prompts and performs a heuristic search guided by multi-attribute feedback. 2) The Dynamic Contrastive Guidance Update (DCGU) mechanism dynamically adjusts the search direction by contrasting the semantic features of high-quality and low-quality prompts.

**Strengths:**

* The framework is a zero-order, black-box optimization framework that requires no gradient information. This makes it universally applicable to any Text-to-Image (T2I) model.
* The paper's approach of embedding an LLM into a classic heuristic search framework (such as PSO/GA) is insightful.
* The proposed DCGU mechanism provides a clear and effective method for escaping local optima by learning from "good" and "bad" features from historical iterations and injecting this knowledge as additional guidance into subsequent searches.

**Weaknesses:**

* The core of DPE-MAF is essentially using an LLM to generate a candidate set, while the introduced DCGU mechanism is fundamentally a form of contrastive learning. This raises concerns about the incremental nature of the improvement.
* Over-reliance on a black box and a reproducibility crisis. The entire DPE-MAF framework is built upon one or more unspecified LLMs. The authors fail to mention which specific LLM was used (GPT-3.5? GPT-4? Llama?), nor do they provide version numbers or API parameters (e.g., temperature). This is a critical oversight. It is well-known that different LLMs, or even different versions of the same LLM, perform vastly differently on the same natural language instructions. This means the paper's results are difficult to reproduce. If a different LLM were used, the behavior and final performance of the entire "algorithm" could change dramatically. For a paper claiming to be published at ICLR, the reproducibility of its core method should not be so fragile.
* Lack of crucial experiments. Since different LLMs could lead to different performance, the paper lacks an ablation study on the impact of using various LLMs.
* The method's reliance on an LLM means the same input can lead to different outputs, introducing significant randomness. From the perspective of rigor and experimental completeness, the authors should have reported standard deviations across different random seeds.
* Contradictory statement on LLM usage. The appendix states that the LLM was only used for "polishing," but in fact, the paper uses the LLM as the very heart of its method.
* The paper claims the optimization process adheres to the constraint of preserving core semantics. However, this is executed merely by being "instructed to ‘preserve the core semantics...’." This is an extremely weak and unverifiable constraint. Semantic drift is inevitable when LLMs rewrite sentences.
* Although the paper provides quantitative evaluations, these metrics struggle to effectively assess alignment with human aesthetics, particularly because the LLM is optimized based on these very quantitative scores. A user study is notably absent.
* This leads to a more fundamental question: Is the reported significant performance boost (especially in CLIPScore) due to finding a "better expression" of the same semantics, or because the LLM "optimized" the original prompt into a new one that is semantically slightly different but easier for the diffusion model to understand and render? For instance, optimizing "a dog sitting on a motorcycle" to "a high-resolution photo of a golden retriever perched calmly on the seat of a parked Harley-Davidson motorcycle." The latter may indeed produce a better image, but it has altered the original intent by adding information. The authors provide no mechanism to quantify or control this semantic drift, making the foundation of their core argument—"optimizing prompt expression"—unstable. Is this "optimization" or "guided re-creation"?

**Questions:**

* Which specific LLM was used? This is crucial for reproducibility, and the authors should provide a reasonable justification for their choice.
* How does the LLM respond to different types of feedback? It would be beneficial to see a curve showing the change in a metric as the feedback is iterated, which would clarify how the LLM's strategy changes over time and what its impact is.
* Is the reported significant performance boost a result of "optimization" or "guided re-creation"?

---

### Official Review · Reviewer_Lgcy · 2025-10-31

**Soundness:** 2
**Presentation:** 3
**Contribution:** 3
**Rating:** 4
**Confidence:** 4

**Summary:**

This paper proposes Dynamic Prompt Evolution via Multi-Attribute Feedback (DPE-MAF), a new framework for prompt optimization in text-to-image generation. Unlike prior works that mainly adjust prompt strength or fine-tune diffusion models, DPE-MAF focuses on overcoming the expressive limitations of prompts themselves.

**Strengths:**

The paper is easy to follow and communicates its ideas effectively.

The paper treats prompt optimization as a zero-order optimization problem and proposes a distinctive method leveraging LLM prior knowledge.

**Weaknesses:**

- Unfair comparison.
In Table 1, Promptist and PAE are trained on user-provided prompts built on SD 1.4. Applying them to SDXL reduces their effectiveness, since the prompt distributions they learn differ from that of SDXL’s training set. Are there any comparison results on SD1.4?
In Tables 1 and 2, the results of the base model are not reported. For example, in Table 1, what are the results of directly generating with SDXL?


- What is the time cost of each processing module? How does it compare with Promptist, PAE, and other baselines?

- Some experimental details are unclear. For instance, which specific LLM was used? How stable and reliable are its outputs? Are there cases of incoherent or nonsensical generations? Since the proposed method relies heavily on the capability of the LLM, reporting these details would make the paper stronger.

- The related work section could be improved by discussing and comparing with recent studies such as:

Hard Prompts Made Easy: Gradient-based Discrete Optimization for Prompt Tuning and Discovery, NeurIPS 2023.

ReNO: Enhancing One-step Text-to-Image Models through Reward-based Noise Optimization, NeurIPS 2024.

**Questions:**

Could you provide some examples of the F+ and F- values extracted by the model in the Dynamic Contrastive Guidance Update section, such as the example in Figure 4? And how exactly do they affect the generated results?

---

### Note · Authors · 2025-11-28

I have read and agree with the venue's withdrawal policy on behalf of myself and my co-authors.